# Citizens’ Literacy in Genomics: A Delphi Survey of Multidisciplinary Experts in the Field

**DOI:** 10.3390/genes13030498

**Published:** 2022-03-11

**Authors:** Giovanna Elisa Calabrò, Michele Sassano, Stefania Boccia

**Affiliations:** 1Section of Hygiene, University Department of Life Sciences and Public Health, Università Cattolica del Sacro Cuore, 00168 Rome, Italy; michele.sassano02@icatt.it (M.S.); Stefania.Boccia@unicatt.it (S.B.); 2Department of Medical and Surgical Sciences, University of Bologna, 40126 Bologna, Italy; 3Department of Woman and Child Health and Public Health-Public Health Area, Fondazione Policlinico Universitario A. Gemelli IRCCS, 00168 Roma, Italy

**Keywords:** genomics, omics sciences, citizens’ literacy, Delphi survey, healthcare experts, precision health

## Abstract

Introduction: Citizens’ literacy in the field of genomics represents one of the cornerstones of proper implementation of genomics in healthcare services. In order to identify the most effective tools by which to elevate citizens’ literacy in genomics, we conducted a survey among the group of multidisciplinary experts within the Italian Network of Genomics in Public Health (GENISAP). Methods. Two rounds of Delphi surveys were carried out in order to identify the main topics, tools, settings, and healthcare professionals’ backgrounds that might usefully be included in citizens’ training initiatives in genomics. To this end, we distributed a questionnaire with 39 items that are scored on a 5-point scale. Results. By the end of the Delphi process, 43 items were selected (19 for the topics, 6 for the tools, 9 for settings, and 9 for the healthcare professionals’ backgrounds). Genomic tests and counseling were among the main topics included, while in terms of tools, face-to-face discussion with healthcare professionals was prioritized by the experts. Among the most appropriate platforms, the group suggested internet/social media and healthcare settings. The healthcare professional considered to have the most relevant role in terms of citizens’ education was a medical doctor with a specialism in clinical genetics. Discussion. Our study attempted to identify the main characteristics that could guide the design of interventions to promote public literacy regarding the field of genomics. Specifically, we have identified the main topics to be included in an educational program for citizens, the tools and settings to consider when providing educational initiatives on genomics, and the healthcare professionals who need to be involved in these initiatives. Therefore, the results of our study provide the necessary basis for the realization of new training initiatives on genomics to be proposed and offered to citizens, these initiatives to be implemented at a national and international level for achieving the transformational change in health systems that is required by the precision health approach.

## 1. Introduction

With the completion of the sequencing of the human genome in 2003, the traditional approach to disease management has been progressively replaced by an individualized approach, driven by advances in the field of “omics” sciences. This new model of personalized healthcare has important implications for all stakeholders, including citizens, healthcare professionals, and policymakers [1]. Thus, the correct implementation of precision health (PH) into practice depends not only on each country’s capability and capacity to support an increasing application of genomics and omics technologies in clinical practice but also on their ability to improve citizens’ involvement and literacy in the field of genomics and other omics sciences [2].

The past two decades have been characterized by an omics revolution, one that began with genomics and today includes a wide range of disciplines that study pools of biological molecules with different functions within living organisms [3]. Some examples are, therefore, genomics, which studies the genes contained in DNA and their multiple functions; transcriptomics, which investigates the DNA transcription product (RNA); proteomics, which studies the proteins encoded by DNA through RNA; metabolomics, which analyzes the molecules that interact within an organism or metabolites; microbiomics, which studies the connections and reciprocal interactions between the pool of biological molecules and microorganisms of the intestinal flora; and nutribiomics, which explores the links between biological molecules and foods and/or nutrients [2].

An important contribution of omics research is to provide more and new information that can transform healthcare by ensuring early diagnoses, more effective prevention programs, and greater precision in the treatment of diseases [4].

Citizens’ empowerment in understanding omics sciences is an essential driver for achieving the transformational change in health systems that is required by the PH approach [4]. Indeed, the successful integration of omics data into healthcare also depends on citizens’ education and awareness, since they are expected to adopt new behaviors in relation to their own health [5,6].

The last few years have been characterized by a disruptive availability of genetics and genomics tests, and advances in the field of omics sciences have had an important impact on public health, not only in terms of interesting benefits but also potential risks for the citizens [6].

In this context, it is worth mentioning direct-to-consumer genetic tests (DTC-GTs); these are tests sold over the internet directly to consumers, without counseling by a healthcare professional. In addition to discriminatory and privacy issues, and the companies not always being adequately regulated [7], citizens do not have the skills needed to understand the results of these tests, possibly leading to cascading and unnecessary diagnostic investigations with the consequent waste of health resources [8].

Therefore, on the one hand, the involvement of trained healthcare professionals in the genetic testing process is crucial; on the other hand, a population that is adequately literate in the omics sciences and related new technologies is needed [2].

In order to identify the main elements that should be included in citizens’ educational initiatives on genomics and the most effective methods for their delivery, we carried out a survey within a multidisciplinary group of experts belonging to the Italian Network of Genomics in Public Health (GENISAP).

## 2. Materials and Methods

### Delphi Process

Forty-five healthcare experts from the Italian Network of GENISAP were involved in a two-round Delphi survey. They did not receive any compensation for their participation. Delphi is a method used to reach, though an iterative process, a consensus on a specific topic among experts with knowledge and experience in that area [9]. GENISAP is a multidisciplinary group of experts, composed of Italian professionals with extensive experience and expertise in the field of genomics, with the role of acting as a scientific advisory group to the Italian Ministry of Health for issues related to genomics and omics sciences and to the implementation of related health policies in the Italian context [10]. The network includes 24 geneticists, 15 public health experts, 1 bioethicist, 2 oncologists, and 3 experts in the economics and management of health services.

Published evidence on citizens’ knowledge in omics science, and related educational initiatives [2,11], were used by the investigative team for the development of an ad hoc questionnaire.

The first version of the questionnaire, administered during the first round of the Delphi process, included 4 sections. In the first section, experts were asked to report their basic personal and professional information, including age, gender, professional category, length of work experience in the field of genomics and omics sciences, and teaching experience in the field of genetics/omics. The next sections included those items to be rated according to their relevance to the Delphi process. To be specific, 14 items were related to the topics, 7 items to the tools, 12 to the settings, while 6 items were for the healthcare professionals involved in educational initiatives aimed at the citizens.

The experts were invited to participate via an invitation e-mail introducing the objective of the study and the content of the Delphi process. After the invitation to participate, a reminder for each round was sent out, aiming at increasing the response rate. The experts were asked to anonymously rate the importance of each proposed questionnaire item through a 5-point scale and to propose new items and amendments to the existing ones if deemed necessary. In the first round, items that received a mean score of ≤2 were excluded; those with a mean score of >2 were proposed again in the second round. Such a cut-off was chosen arbitrarily. Similarly, the amendments and new items proposed by experts during the first round were proposed for other experts’ evaluation during the second round. The threshold for exclusion of the items in the second round was more restrictive than the one used for the first round, being represented by a mean score of <3, in order to identify the essential items for the training of citizens in genomics. Figure 1 reports a flowchart of the Delphi process.

## 3. Results

During the first and the second rounds of the Delphi process, 32 (71.11% of the 45 invited experts) and 19 (42.22%) GENISAP members participated, respectively.

The participants of the first round had a mean age of 55.69 years (standard deviation, SD: 9.94), half (50%) of them were male and half (50%) female. Overall, 34.38% (*n* = 11) of the participants were represented by biologists, 21.88% (*n* = 7) by public health physicians and other public health officials, 18.75% (*n* = 6) by medical geneticists, 15.62% (*n* = 5) by other specialist physicians, 6.25% (*n* = 2) by biotechnologists, and 3.13% (*n* = 1) by economists. Most of them reported previous professional experience in the field of genetics or omics sciences over a period of more than 10 years (68.75%), along with previous experience in teaching genetics or omics sciences (68.75%). Among those who stated they had previous experience in teaching, this had lasted for over 10 years for most of them (63.64%). Similarly, the participants of the second round, with a mean age of 56.00 years (SD: 10.34), were 47.37% male and 52.63% female. Among them, 36.84% (*n* = 7) were biologists, 36.84% (*n* = 7) were medical geneticists, 21.05% (*n* = 4) were public health physicians and other public health officials, and 5.26% (*n* = 1) were biotechnologists. In parallel with the results of the first round, most of the participants stated that they had previous professional experience in the field of genetics or omics sciences for a period of more than 10 years (73.68%) and previous experience in teaching genetics or omics sciences (78.95%). For 66.67% of those who reported having previous experience in the field of teaching, this was for a period of over 10 years.

A flowchart of the results of the Delphi process is shown in Figure 2.

Among the 39 items proposed in the first round (Table 1), none were excluded by the participants. Therefore, all items were presented again in the second round.

Furthermore, 13 additional items (6 for topics, 1 for tools, 2 for settings, and 4 for professionals) were proposed by the participants in the first round and were, thus, presented in the second round (Table 2).

Of the 52 items (20 for the topics, 8 for the tools, 14 for the settings, and 10 for the healthcare professionals’ backgrounds) presented in the second round, 43 were finally selected by the participants.

Among the topics to address in terms of the citizens’ education, 19 items were included that can be grouped into five macro-areas, such as the basic concepts of genetics and heredity (*n* = 1), prevention (*n* = 3), scientific research (*n* = 2), the role of omics sciences in specific fields (*n* = 6), and genetic/genomic tests (*n* = 7). The details of the topics included, grouped into macro-areas, are shown in Table 3.

Regarding the tools deemed adequate for training and informing citizens, 6 items were included, such as discussion groups with healthcare professionals or with teachers for school-aged individuals, audiovisual media, ad hoc information campaigns, news bulletins, and articles in newspapers and periodicals (Table 4).

Furthermore, 9 items were included among the settings that were deemed appropriate as places and contexts wherein to carry out citizens’ education activities, including the internet and social media, health and health-related settings, schools, research centers, and cultural and educational centers such as museums, libraries, zoos and aquariums (Table 5).

Finally, regarding the most appropriate professionals to involve, greater importance was given to geneticists, both doctors and biologists, and for genetic counselors, followed by public health physicians, sociologists, and communication experts (Table 6).

## 4. Discussion

The results of the Delphi process summarize experts’ opinions on the elements that should be included in training activities aimed at educating citizens in the field of genomics, along with the most effective methodologies and tools for achieving this goal.

Among the topics that the experts highlighted for the education of citizens, five macro-areas related to genomics and other omics sciences emerged, these being: the basic concepts of genetics and heredity; prevention; scientific research; the role of omics sciences in specific fields, and their specific applications; and genetic/genomic tests.

According to the opinion of the GENISAP experts, fundamental elements in citizens’ educational process are related to the basic concepts of genetics and heredity, with reference to the structure and function of DNA, the general principles of heredity, the burden of genetic diseases, the significance of chromosomal and complex diseases, and the components of multifactoriality [12]. Other relevant topics for citizens’ education include the dissemination of basic knowledge on the application of genomics at all levels (primary, secondary and tertiary) of prevention, paying particular attention to the effect of genetic and non-genetic risk factors on the onset of diseases and to disease prevention approaches before conception, during pregnancy and in the first years of life. Furthermore, to ensure the greater empowerment of the citizens, it will be necessary to educate them on the role and application of omics sciences in specific fields (e.g., oncology, aging, cardiovascular diseases, forensics), on the disruptive innovation related to gene therapy, and on important advances in the field of nutrigenomics, epigenetics and the influence of microbiota and the microbiome on health. An adequate degree of knowledge on genetic/genomic tests, related technologies, and their applications was also deemed by experts as important for the citizens with the need to underline not only their benefits but also their limitations and potential risks, which are mainly linked to the use of DTC-GTs and to the circulation on the internet of fake news about genetic testing—as well as information about their accessibility and their costs. However, in order to educate the public on these issues, and in particular on the benefits and limits of DTC-GTs, it will be necessary to adequately and preliminarily train healthcare professionals. Eventually, the public should also be educated on scientific advances in genomic research and on the ethical implications related to omics research and the use of genetic/genomic tests.

As for the tools to deliver educational initiatives to citizens, the experts indicated that face-to-face discussions with healthcare professionals and with teachers were a priority for school-aged individuals.

Among the most appropriate settings, experts suggested the internet/social media, healthcare settings, and schools. The professionals considered that the most relevant people to be involved in citizens’ education were medical and clinical geneticists.

Public standards of literacy in the omics field represent a fundamental public health intervention to be implemented to enable adequate health decisions by citizens. Among the topics that experts considered relevant for citizens’ education, basic concepts of genetics and heredity emerged, as well as more focused concepts (e.g., genetic/genomic tests and their related issues and limitations, including the DTC-GTs). These results are in line with the findings of our previous systematic review [2], where genetic/genomic research, disease etiology and susceptibility, genetic and omics tests were among the main topics identified as being relevant for citizens’ education. However, previously implemented educational initiatives on genetics/genomics aimed at informing the public did not always focus on such topics [11]. Thus, it might be important to align future initiatives with real citizens’ educational needs and wishes. This, indeed, might also represent a strategy by which to foster citizens’ interest and, hence, their participation in this kind of initiative. In our Delphi survey, proper citizens’ education as a means to tackle the spread of fake news was considered to be relevant, according to the experts. The internet has become a popular resource for learning about health topics. However, given the large amount of inaccurate information that can be found online, citizens can easily be misinformed [13]. This underlines the importance of the need for improving citizens’ literacy and for reliable information sources, especially in complex fields such as genomics and other omics sciences.

As regards the tools that were deemed adequate for training and informing citizens, both the methods based on mass media (e.g., movies and videos) and on more traditional means (e.g., lessons and discussion groups) were selected by experts, even though the former scored higher than the latter. This parallels the landscape of educational initiatives carried out worldwide, which has been highly heterogeneous in this sense [11]. However, according to the survey results, one of the tools deemed most appropriate for citizens’ literacy by experts is represented by discussion with healthcare professionals, although the composition of the Delphi survey participants might have influenced their prioritization of discussions with healthcare professionals as a tool for promoting citizens’ genomic literacy. Our previous studies [2,11] have, instead, reported that the main sources of information for citizens are TV, magazines, newspapers, and the internet. These data, associated with the low level of citizens’ knowledge in the omics sciences field, further underlines the healthcare professional’s fundamental role in the education of the general population. In order to address this role appropriately, healthcare professionals should, in turn, be adequately trained regarding these issues [5,14,15], as their insufficient skills or inadequate attitudes could represent an obstacle to the development of omics sciences and the effective implementation of personalized healthcare. For this reason, this aspect has indeed been addressed by decision-makers and public health experts [5].

Among the settings deemed functional for the improvement of citizens’ literacy, the list comprised various items, including the internet and social media, healthcare settings (in particular, primary care settings, but also alternative ones, such as pharmacies), schools, research centers, and cultural centers (e.g., libraries and science museums). Research and cultural centers are useful sources of informal learning for high schools or upper schools attempting to teach young people about genetics [11], although typically, incorporation into their program requires outside funding and other support. Their placement in the bottom two rows of Table 5 reflects this underutilization. However, these two avenues represent available educational resources that are as yet often untapped. Most of the educational initiatives conducted to date on omics sciences, aimed at citizens, were web-based resources [11]. Given the considerable development and growth in the use of the internet and social networks over recent years, it is also necessary to consider these means of information for citizens’ literacy, as they are able to reach younger people, in particular [16]. However, based on the results of the survey, the use of digital means needs to be strengthened to raise public awareness, but without neglecting the importance of events that involve in-person attendance.

Eventually, according to the data that emerged from our survey with the experts, the healthcare professional considered to be the most important in citizens’ education was the medical geneticist, followed by the genetic biologist, the genetic counselor, and the public health physician. An interesting result is that medical specialists (e.g., oncologists, cardiologists, gynecologists, neurologists, etc.) were rated by GENISAP experts in the second to last position. Usually, medical specialists familiar with and utilizing genetic services are viewed as experts in the use of genetic testing. Probably, as in the case of general practitioners, this position depends on the fact that not all specialists can be considered as “educators” of citizens in the field of genomics and omics sciences if they are not properly trained in these fields.

However, the results of the survey also highlight the importance of other professionals, such as nurses with specific training in the genetics/genomics field, sociologists, and communication experts, if properly trained. The professionals to be involved also include general practitioners who are in closer contact with citizens and patients, if they are adequately updated on the topics. The diversity of the healthcare professionals’ backgrounds as identified by the GENISAP experts for citizens’ education underlines the multidisciplinarity that characterizes the complex world of genomics and omics sciences, as well as the contribution that each professional can make in the process of improving citizens’ literacy.

To the best of our knowledge, this is the first study to identify the main characteristics that could guide the design of interventions to promote citizens’ literacy in the omics field. Specifically, we have identified the main topics to be included in an educational program for citizens, the tools and settings to be considered when providing educational initiatives, and the healthcare professionals who need to be involved in these initiatives. However, our paper has some limitations. The most important one is that the items were identified based only on Italian experts’ opinions. Therefore, items that were included in the final Delphi round might not be applicable to other contexts. Furthermore, the experts were members of an Italian network for genomics in public health, belonging to different health disciplines, and differently represented within the network. However, the network did not include general practitioners, who are in close contact with patients and citizens. As regards the sample size, the number of survey participants was high in both the first and second rounds (32 and 19 experts, respectively), with the minimum acceptable number to ensure the validity of a Delphi process being between 8 and 12 experts [17]; furthermore, the experts involved in our study were all experts in the field of genomics and other omics sciences, while non-experts were not involved in any phase of the process. As such, the identified items should also be further integrated by consulting the citizens.

Eventually, the results of our study provide the necessary basis for the development and implementation of new training activities for citizens in the field of genomics and omics sciences, and the suitable methodologies and tools for achieving this goal.

## 5. Conclusions

Improving citizens’ literacy in the field of genomics and omics sciences represents a fundamental public health intervention, to allow the public to make adequate choices regarding their own health. Therefore, the promotion of new training strategies and innovative educational interventions, aimed at the citizens in this field, is needed. To achieve this result evidence-based tools for organizing effective educational strategies are needed. Our work is a first proposal aimed at supporting this important goal.

## Figures and Tables

**Figure 1 genes-13-00498-f001:**
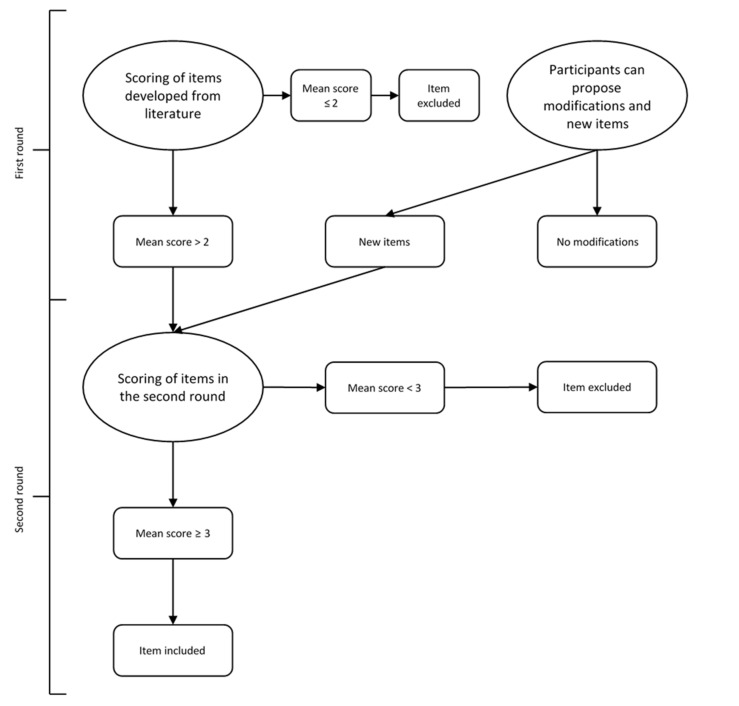
Flowchart of the selection procedure of the Delphi process.

**Figure 2 genes-13-00498-f002:**
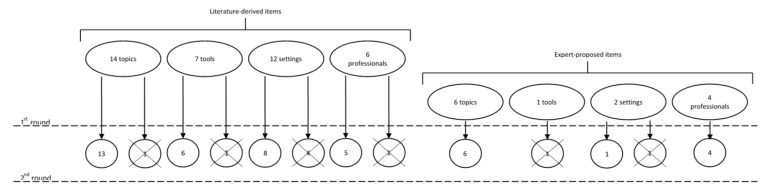
Flowchart of the results of the Delphi process.

**Table 1 genes-13-00498-t001:** Items included in the first version of the survey and the results of the first and second rounds of the Delphi process.

	First Round Mean Score (Out of 5)	Second Round Mean Score (Out of 5)
Topics to address in citizens’ educational initiatives		
Basic concepts of genetics, the structure and function of DNA, general principles of heredity, the burden of genetic diseases, chromosomal disorders, complex diseases, components of multifactoriality	3.84	4.11
Determinants of health and disease: the concept of the influence of genetic and non-genetic risk factors (environmental factors, lifestyle, etc.) on the risk of disease	4.56	4.53
Genetic tests and the main omics technologies currently available, with their application	3.97	4
Pre- and post-test genetic counseling	4.19	4.53
Genetic tests in the prenatal setting	4.13	4.37
Accessibility and costs of genetic/omics tests	3.97	3.84
Direct-to-consumer genetic tests: what they are, validity, their implications for the patient and their regulation; the importance of pre- and post-test counseling	4.44	4.53
Role of omics sciences in specific fields (e.g., oncology, aging, cardiovascular diseases, forensics)	3.88	3.84
Nutrigenomics and related tests, the validity and usefulness of nutrigenomics and personalized nutrition approaches in the prevention of diseases	3.78	3.37
Implications of genetically modified organisms	3.66	**2.89 ***
Influence of microbiota and the microbiome on health	3.75	3.37
Implications and future developments of genomics/omics research	3.59	3.68
Ethical implications related to genetic/omics research and the use of genetic/omics tests	3.97	4.05
Fake news about genetic testing	4.44	4.42
**Tools to deliver citizens’ educational initiatives**		
Films, short films, and videos, including through specific websites (e.g., YouTube)	4.16	4
Lessons and discussions with teachers for school-aged individuals	4.19	4.11
News bulletins	3.16	3.21
Information campaigns	4.03	3.89
Reading books, articles, newspapers, magazines	3.38	3.16
Discussion groups and focus groups with healthcare professionals	3.81	4.11
Discussions with family and/or friends	2.47	**2.58**
**Settings for citizens’ educational initiatives**		
Internet	4.13	4.37
Social media	3.97	3.95
TV and radio	3.78	3.63
School	4.56	4.11
Libraries, science museums, zoos, aquariums	3.38	3.05
Research centers	3.59	3.37
Public places (e.g., squares, information stands, shopping centers)	3.19	**2.74**
Recreational places (e.g., cafes and bistros)	2.75	**2.53**
Healthcare settings	3.94	4.26
Alternative healthcare settings (pharmacy, dental offices, and similar)	3.59	3.63
Religious institutions	2.38	**2.21**
Home environment	2.31	**2.26**
**Professionals to involve in citizens’ educational initiatives**		
Clinical geneticists	4.38	4.47
Specialist physicians (e.g., oncologist, cardiologist, gynecologist, neurologist, etc.)	3.78	3.05
Public health physicians	3.93	3.68
General practitioners	3.72	3
Nurses	3.03	**2.42**
Biologists	3.69	3.32

* Scores lower than the predefined threshold for exclusion (< 3 for the second round) are reported in bold.

**Table 2 genes-13-00498-t002:** New items proposed in the first round and results of the second round of the Delphi process.

	Second Round Mean Score (out of 5)
Topics to address in citizens’ educational initiatives	
Limitations of genetic tests, difficulty in interpretation of results, the importance of performing the omics test in a multidisciplinary context, the possibility of non-conclusive test results	4.11
Prevention: definition of prevention, primary and secondary prevention, organized screening, costs/benefits	4.16
Ability of the epigenome to respond to environmental exposure and lifestyle	3.37
New approaches for disease prevention, with a focus on disease prevention before conception, during pregnancy, and in the first years of life	3.58
Interaction and synergies between genetic and environmental risk factors, epigenetic mechanisms	3.53
New gene therapies	3.58
**Tools to deliver citizens’ educational initiatives**	
Newsletters and similar tools	**2.84 ***
**Settings for citizens’ educational initiatives**	
Primary care settings	4
Cinemas	**2.26**
**Professionals to involve in citizens’ educational initiatives**	
Sociologists and specifically trained communication experts	3.47
Genetic counselor	3.89
Genetic biologists	4.05
Specifically trained nurses	3.42

* Scores lower than the predefined threshold for exclusion (<3) are reported in bold.

**Table 3 genes-13-00498-t003:** Items selected in the category “Topics to address in citizens’ educational initiatives”, according to macro-area (within each macro-area, items are listed by decreasing mean score).

Topics	Macro-Area
Basic concepts of genetics, the structure and function of DNA, general principles of heredity, the burden of genetic diseases, chromosomal disorders, complex diseases, components of multifactoriality	**Basic concepts of genetics and heredity**
Determinants of health and disease: the concept of genetic and non-genetic risk factors (environmental factors, lifestyle, etc.) on the risk of disease	**Prevention**
Prevention: definition of prevention, primary and secondary prevention, organized screening, costs/benefits	
New approaches for disease prevention, with a focus on disease prevention before conception, during pregnancy, and in the first years of life	
Ethical implications related to genetic/omics research and the use of genetic/omics tests	**Scientific research**
Implications and future developments of genomics/omics research	
Role of omics sciences in specific fields (e.g., oncology, aging, cardiovascular diseases, forensics)	**Role of omics sciences in specific fields and specific applications of omics sciences**
New gene therapies	
Interaction and synergies between genetic and environmental risk factors, epigenetic mechanisms	
Nutrigenomics and related tests, the validity and usefulness of nutrigenomics and personalized nutrition approaches in the prevention of diseases	
Influence of microbiota and the microbiome on health	
Ability of the epigenome to respond to environmental exposure and lifestyle	
Pre- and post-test genetic counseling	**Genetic/genomic tests**
Direct-to-consumer genetic tests: what they are, their validity, implications for the patient, and their regulation; the importance of pre- and post-test counseling	
Fake news about genetic testing	
Genetic tests in the prenatal setting	
Limitations of genetic tests, difficulty in interpretation of results, the importance of performing the omics test in a multidisciplinary context, the possibility of inconclusive test results	
Genetic tests and main omics technologies that are currently available and their application	
Accessibility and costs of genetic/omics tests.	

**Table 4 genes-13-00498-t004:** Items selected in the category “Tools to deliver citizens’ educational initiatives“, listed by decreasing mean scores from the Delphi process.

Tools
Discussion groups and focus groups with healthcare professionals
Lessons and discussions with teachers for school-aged individuals
Films, short films and videos, including via specific websites (e.g., YouTube)
Information campaigns
News bulletins
Reading books, articles, newspapers, magazines

**Table 5 genes-13-00498-t005:** Items selected in the category “Settings for citizens’ educational initiatives”, listed by decreasing mean scores from the Delphi process.

Settings
Internet
Healthcare settings
School
Primary care settings
Social media
TV and radio
Alternative healthcare settings (pharmacy, dental office, and similar)
Research centers
Libraries, science museums, zoos, aquariums

**Table 6 genes-13-00498-t006:** Items selected in the category “Professionals to involve in citizens’ educational initiatives”, listed by decreasing mean scores from the Delphi process.

Professionals
Medical and clinical geneticists
Genetic biologists
Genetic counselors
Public health physicians
Sociologists and specifically trained communication experts
Specifically trained nurses
Biologists (not geneticists)
Specialist physicians (e.g., oncologist, cardiologist, gynecologist, neurologist, etc.)
General practitioners

## Data Availability

Not applicable.

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
