# Peer review of "Citizens’ Literacy in Genomics: A Delphi Survey of Multidisciplinary Experts in the Field"

_genes, 2022, doi:10.3390/genes13030498_

Round 1
Reviewer 1 Report
The number of items and topics to be addressed resulting from the arbitrary cut-off in Round 1 of less than or equal to 2 means the subsequent round with a cut-off of 3 did not provide a realistic content or settings for any educational program. Public (citizen) education requires targeted and focussed and realistic goals. While the results may inform a citizen consultation, the discussion should address those that were of highest priority. Further, some further discussion is warranted of the limited expertise of health professionals in omics and therefore their education and training strategies may need ot be undertaken before any citizenry education initiatives given the emphasis on their roles identified.
Author Response
Dear Editor, Dear Reviewer,
We would like to thank you for your valuable comments and for the opportunity to resubmit our work. We have amended the paper according to the received suggestions and we hope that it now appears improved.
Hereafter point-to-point answers are provided.
Best regards,
Giovanna Elisa Calabrò
REVIEWER 1
Comment 1: The number of items and topics to be addressed resulting from the arbitrary cut-off in Round 1 of less than or equal to 2 means the subsequent round with a cut-off of 3 did not provide a realistic content or settings for any educational program. Public (citizen) education requires targeted and focussed and realistic goals. While the results may inform a citizen consultation, the discussion should address those that were of highest priority.
Answer: We thank the Reviewer for his/her important comments. We agree with the reviewer on the limits related to the arbitrary choice of the cut off proposed for the selection of items in I and II round of our Delphi and we are aware that this is not the only process to be considered for the construction of an educational program for citizens. In fact, we reported in discussion that "the identified items should be further integrated or validated by consulting citizens too”.
However, we would like to specify that the objective of our study was to identify - on the basis of the literature evidence used for the construction of the questionnaire, and the opinion of genomics experts - the main topics, tools, settings and healthcare professionals' backgrounds that might be included in citizens' training initiatives in genomics field.
As regards the cut off selected in the first and second rounds, we have identified a more stringent cut off in the second round precisely to identify the main topics on which to train citizens. However, since the items are numerous, we made some changes in the text by eliminating the term "priority". With our study we identified, with the support of experts, the main characteristics to be considered for the realization of a training initiative on genomics aimed at citizens. Therefore, we also changed the title of the paper.
Comment 2: Further, some further discussion is warranted of the limited expertise of health professionals in omics and therefore their education and training strategies may need ot be undertaken before any citizenry education initiatives given the emphasis on their roles identified.
Answer: We thank the Reviewer for his/her important comment. We agree with the Reviewer: we need professionals trained in the omics sciences. In fact, we underlined this aspect both in the introduction ("the involvement of trained healthcare professionals in the genetic testing process is crucial") and in the discussion of the paper ("These data associated with the low level of citizens' knowledge in the omics sciences field further underlines the healthcare professionals' fundamental role in the education of the general population. In order to address this role appropriately, healthcare professionals should in turn be adequately trained on these issues [5, 132, 143], as their insufficient skills or inadequate attitudes could represent an obstacle to the development of omics sciences and the effective implementation of personalized healthcare. For this reason, this aspect has been indeed addressed by decision makers and public health experts [5] ").

Reviewer 2 Report
General Comments to the Authors:
This article seems timely given the increasing importance of precision health and the author’s previous systematic review work in this area. Findings are equally relevant to medical and public health practitioners. The tables suggest some additional considerations I list below that could be added to the Discussion section. Findings were somewhat predictable given the expert group approached, but are nonetheless significant. Abstract plus a few other lines in the narrative need grammatical updating.
Specific Comments to the Authors:
Abstract –
P. 1 –
line 14: in the healthcare services. -> in healthcare services.
lines 16-17: Two-round of Delphi surveys -> Two rounds of Delphi surveys
lines 18, 21: healthcare professionals’ figures -> healthcare professionals’ backgrounds
line 23: Inter-net -> Internet
lines 31-32: implemented at national and international level -> implemented at a national and international level
line 32: trans-formational -> transformational
Narrative –
P. 2, lines 74, 5 –
Published evidences -> Published evidence -or- Published sources of evidence
were used for the development -> were used by the investigative team for the development
P. 3, line 107 –
Be sure to check whether the n=7 public health physicians was actually n=7 public health physicians and other public health officials
P. 3, line 114 –
“mean age of 56 years” – Why is this figure rounded yet the mean age for the first round has two decimal points included (55.69)?
P. 4, line 125 –
“in the first round (table 2).”: Do you mean “in the second round (table 2)”?
Given the extensive list of items dealt with, you might mention whether participants received any form of compensation.
P. 8, line 173 –
Mention of genetic diseases, common, complex diseases, and multifactoriality calls for a citation.
P. 8, line 184 –
DTC-GTs deemed as an important topic by the experts:
Yet DTC genetic tests as a topic and limitations of genetic tests had separate ratings in the Genetic/genomics tests category (Table 3). Please consider adding a comment about the need to educate healthcare professionals in the limitations of DTC genetic testing, perhaps placing such a comment before the sentence on educating citizens (line 187).
P. 8, line 191 or P. 9, line 220 –
Please consider adding a line on whether the composition of the Delphi survey participants might have influenced their prioritization of discussions with healthcare professionals as a tool for promoting citizen genomic literacy.
P. 9, line 233 –
Research centers and cultural centers (e.g., libraries and science museums) as functional settings:
Both types of centers are useful sources of informal learning for high schools or upper schools attempting to teach young people about genetics, though typically their program incorporation requires outside funding and other support. Their placement in the bottom two rows of Table 5 reflects this underutilization. You might mention these two avenues as available yet often untapped educational resources.
P. 9, line 245 –
Medical specialists in the list of professionals able to educate citizens:
Granted they appear as a category in your survey, it is nonetheless fascinating that they were rated by participants in the second to last position. Usually medical specialists familiar with and utilizing genetic services are viewed as experts in the use of genetic testing. Why were they rated so low by this group? Is it because they miss the preventive outlook mentioned in Table 3, have too little time to act as educators, are not formally trained as health educators, …? Could you please add a sentence on their relative placement within Table 6?
P. 9, lines 250 and 262 –
Remove hyphen from “educa-tion” and “dif-ferently”

Author Response
Dear Editor, Dear Reviewer,
We would like to thank you for your valuable comments and for the opportunity to resubmit our work. We have amended the paper according to the received suggestions and we hope that it now appears improved.
Hereafter point-to-point answers are provided.
Best regards,
Giovanna Elisa Calabrò
REVIEWER 2
General Comments:
Comment 1: This article seems timely given the increasing importance of precision health and the author’s previous systematic review work in this area. Findings are equally relevant to medical and public health practitioners. The tables suggest some additional considerations I list below that could be added to the Discussion section. Findings were somewhat predictable given the expert group approached, but are nonetheless significant. Abstract plus a few other lines in the narrative need grammatical updating.
Answer: We thank the Reviewer for his/her comments and suggestions. We made all the required changes and additions. We responded to his/her requests in line by line comments.
Specific Comments to the Authors:
Abstract –
- 1 –
line 14: in the healthcare services. -> in healthcare services.
Answer: We made the required corrections.
lines 16-17: Two-round of Delphi surveys -> Two rounds of Delphi surveys
Answer: We made the required corrections.
lines 18, 21: healthcare professionals’ figures -> healthcare professionals’ backgrounds
Answer: We made the required corrections.
line 23: Inter-net -> Internet
Answer: We made the required corrections.
lines 31-32: implemented at national and international level -> implemented at a national and international level
Answer: We made the required corrections.
line 32: trans-formational -> transformational
Answer: We made the required corrections.
Narrative –
- 2, lines 74, 5 –
Published evidences -> Published evidence -or- Published sources of evidence
Answer: We made the required corrections.
were used for the development -> were used by the investigative team for the development
Answer: We made the required corrections.
- 3, line 107 –
Be sure to check whether the n=7 public health physicians was actually n=7 public health physicians and other public health officials
Answer: We thank the Reviewer for his/her suggestion. We made the required corrections.
- 3, line 114 –
“mean age of 56 years” – Why is this figure rounded yet the mean age for the first round has two decimal points included (55.69)?
Answer: We thank the Reviewer for his/her question. The mean age of participants in the second round was exactly 56.00 years, we have thus added the two decimal points in the text.
- 4, line 125 –
“in the first round (table 2).”: Do you mean “in the second round (table 2)”?
Answer: We thank the Reviewer for his/her suggestion. We made the required corrections.
Given the extensive list of items dealt with, you might mention whether participants received any form of compensation.
Answer: We thank the Reviewer for his/her suggestion. We specified this in the methods section.
- 8, line 173 –
Mention of genetic diseases, common, complex diseases, and multifactoriality calls for a citation.
Answer: We thank the Reviewer for his/her suggestion. We have inserted a new reference (Genetic Alliance; The New York-Mid-Atlantic Consortium for Genetic and Newborn Screening Services. Under-standing Genetics: A New York, Mid-Atlantic Guide for Patients and Health Professionals. Chapter 1 (Genetics 101). Washington (DC): Genetic Alliance; 2009 Jul 8).
- 8, line 184 –
DTC-GTs deemed as an important topic by the experts:
Yet DTC genetic tests as a topic and limitations of genetic tests had separate ratings in the Genetic/genomics tests category (Table 3). Please consider adding a comment about the need to educate healthcare professionals in the limitations of DTC genetic testing, perhaps placing such a comment before the sentence on educating citizens (line 187).
Answer: We thank the Reviewer for his/her comments and suggestions. We added the following sentence “However, in order to train citizens on these issues, and in particular on the benefits and limits of DTC-GTs, it will be necessary to adequately and preliminarily train healthcare professionals”.
- 8, line 191 or P. 9, line 220 –
Please consider adding a line on whether the composition of the Delphi survey participants might have influenced their prioritization of discussions with healthcare professionals as a tool for promoting citizen genomic literacy.
Answer: We thank the Reviewer for his/her suggestion. We added the following sentence “However, according to survey results, one of the tools deemed more appropriate for citizens' literacy by experts is represented by discussion with healthcare professionals, though the composition of the Delphi survey participants might have influenced their prioritization of discussions with healthcare professionals as a tool for promoting citi-zens genomic literacy”.
- 9, line 233 –
Research centers and cultural centers (e.g., libraries and science museums) as functional settings:
Both types of centers are useful sources of informal learning for high schools or upper schools attempting to teach young people about genetics, though typically their program incorporation requires outside funding and other support. Their placement in the bottom two rows of Table 5 reflects this underutilization. You might mention these two avenues as available yet often untapped educational resources.
Answer: We thank the Reviewer for his/her comment and suggestion. We added the following sentence in the discussion section “Research and cultural centers are useful sources of informal learning for high schools or upper schools attempting to teach young people about genetics [10], though typically their program incorporation requires outside funding and other support. Their placement in the bottom two rows of Table 5 reflects this underutilization. However, these two avenues represent available educational resources yet often untapped”.
- 9, line 245 –
Medical specialists in the list of professionals able to educate citizens:
Granted they appear as a category in your survey, it is nonetheless fascinating that they were rated by participants in the second to last position. Usually medical specialists familiar with and utilizing genetic services are viewed as experts in the use of genetic testing. Why were they rated so low by this group? Is it because they miss the preventive outlook mentioned in Table 3, have too little time to act as educators, are not formally trained as health educators, …? Could you please add a sentence on their relative placement within Table 6?
Answer: We thank the Reviewer for his/her comment and suggestion. We added the following sentence in the discussion section “An interesting result is that medical specialists (e.g. oncologist, cardiologist, gynecol-ogist, neurologist, etc.) were rated by GENISAP experts in the second to last position. Usually medical specialists familiar with and utilizing genetic services are viewed as experts in the use of genetic testing. Probably, as for general practitioners, this position depends on the fact that not all specialists can be considered as “educators” of citizens in the field of genomics and omics sciences if not properly trained in these fields”.
- 9, lines 250 and 262 –
Remove hyphen from “educa-tion” and “dif-ferently”
Answer: We thank the Reviewer for his/her suggestion. We made the required corrections.

Reviewer 3 Report
This is an important study. However, I have some concerns about this paper.
- Overall, this paper is hard to follow as the writing is unclear.
- The introduction is unclear. I thought this study was to be about citizens, so I am unsure why the authors talked about healthcare professionals and DCT genetic tests. The first paragraph is fine, but the remaining paragraphs need to be re-written.
- The authors may need to explain omics sciences in the introduction.
- Methods: I was unclear why items that received a mean score ≤ 2 were excluded in the first round, and items that received a mean score <3 were excluded in the second round. Please share your rationale for this decision.
- Results: It is better tables are embedded in the text, not in separate pages. Also, this section is hard to follow.
- This study has no IRB review. Why?
Author Response
Dear Editor, Dear Reviewer,
We would like to thank you for your valuable comments and for the opportunity to resubmit our work. We have amended the paper according to the received suggestions and we hope that it now appears improved.
Hereafter point-to-point answers are provided.
Best regards,
Giovanna Elisa Calabrò
REVIEWER 3
This is an important study.
Answer: We thank the Reviewer for his/her positive comment.
However, I have some concerns about this paper.
- Overall, this paper is hard to follow as the writing is unclear.
Answer: We thank the Reviewer for his/her comment. We tried to improve the paper by making further changes. We hope that we have adequately followed up on the Reviewer's requests.
2. The introduction is unclear. I thought this study was to be about citizens, so I am unsure why the authors talked about healthcare professionals and DCT genetic tests. The first paragraph is fine, but the remaining paragraphs need to be re-written.
Answer: We thank the Reviewer for his/her comments. We integrated the introduction as required. We would like to specify that the objective of our study was to identify - on the basis of the literature evidence used for the construction of the questionnaire, and the opinion of genomics experts - the main topics, tools, settings and healthcare professionals' backgrounds that might be included in citizens' training initiatives in genomics field.
3.The authors may need to explain omics sciences in the introduction.
Answer: We thank the Reviewer for his/her suggestion. We integrated the introduction as required.
4.Methods: I was unclear why items that received a mean score ≤ 2 were excluded in the first round, and items that received a mean score <3 were excluded in the second round. Please share your rationale for this decision.
Answer: We thank the Reviewer for his/her comment and suggestion. In the methods section we report that cut-off was chosen arbitrarily and that “The threshold for exclusion of the items in the second round was more restrictive than the one used for the first round, being represented by a mean score < 3”. We decided to consider a more stringent cut-off to the second round in order to identify the essential items for the training of citizens in genomics. We specified this reason in the text of the paper.
5.Results: It is better tables are embedded in the text, not in separate pages. Also, this section is hard to follow.
Answer: We thank the Reviewer for his/her comment and suggestion. We followed up on the Reviewer's request. We realize that the tables can hinder the reading of the paper but at the same time we think that the inclusion of the tables is essential for the accountability of the work carried out as well as for sharing complete information on the results obtained from the survey.
6.This study has no IRB review. Why?
Answer: We thank the Reviewer for his/her question. The survey presented in our paper is part of activities that the Italian Network of Genomics in Public Health (GENISAP) carries out as scientific advisor of the Italian Ministry of Health, in the context of research projects promoted by the National Center for Disease Prevention and Control (CCM) of the Ministry of Health, and focused on genomics and omics sciences.
Therefore, an IRB review was not required for this consultation of GENISAP experts. However, we have better specified the role of GENISAP in the text.
